# COVID-19 and Adverse Pregnancy Outcome: A Systematic Review of 104 Cases

**DOI:** 10.3390/jcm9113441

**Published:** 2020-10-26

**Authors:** Ramy Abou Ghayda, Han Li, Keum Hwa Lee, Hee Won Lee, Sung Hwi Hong, Moonsu Kwak, Minwoo Lee, Minjae Kwon, Ai Koyanagi, Andreas Kronbichler, Louis Jacob, Lee Smith, Jae Il Shin

**Affiliations:** 1Department of Global Health and Population, Harvard T.H. Chan School of Public Health, Boston, MA 02115, USA; ramy.aboughayda@gmail.com (R.A.G.); sunghwihong@gmail.com (S.H.H.); 2Division of Urology, Brigham and Women’s Hospital, and Harvard Medical School, Boston, MA 02115, USA; 3Department of Molecular Genetics and Microbiology, Center for Neurogenetics and the Genetics Institute, College of Medicine, University of Florida, Gainesville, FL 32610, USA; lih2@ufl.edu; 4Department of Pediatrics, Yonsei University College of Medicine, Seoul 03722, Korea; AZSAGM@yuhs.ac (K.H.L.); 60923155@naver.com (H.W.L.); anstn0417@naver.com (M.K.); leeminwoo998@gmail.com (M.L.); minjae_kwon@alumni.brown.edu (M.K.); 5Research and Development Unit, Parc Sanitari Sant Joan de Déu, CIBERSAM, Dr. Antoni Pujadas, 42, Sant Boi de Llobregat, 08830 Barcelona, Spain; a.koyanagi@pssjd.org (A.K.); louis.jacob.contacts@gmail.com (L.J.); 6ICREA, Pg. Lluis Companys 23, 08010 Barcelona, Spain; 7Department of Internal Medicine IV, Nephrology and Hypertension, Medical University Innsbruck, 6020 Innsbruck, Austria; Andreas.Kronbichler@i-med.ac.at; 8Faculty of Medicine, University of Versailles Saint-Quentin-en-Yvelines, Montigny-le-Bretonneux, 78180 Versailles, France; 9The Cambridge Centre for Sport and Exercise Sciences, Anglia Ruskin University, Cambridge CB1 1PT, UK; Lee.Smith@anglia.ac.uk

**Keywords:** coronavirus disease 2019, pregnancy, COVID-19, fetal death, neonatal outcomes, preterm birth, SARS-CoV-2, stillbirth, maternal morbidity, maternal mortality, neonatal morbidity, neonatal mortality

## Abstract

(1) Background: Until now, several reports about pregnant women with confirmed coronavirus disease 2019 (COVID-19) have been published. However, there are no comprehensive systematic reviews collecting all case series studies on data regarding adverse pregnancy outcomes, especially association with treatment modalities. (2) Objective: We aimed to synthesize the most up-to-date and relevant available evidence on the outcomes of pregnant women with laboratory-confirmed infection with COVID-19. (3) Methods: PubMed, Scopus, MEDLINE, Google scholar, and Embase were explored for studies and papers regarding pregnant women with COVID-19, including obstetrical, perinatal, and neonatal outcomes and complications published from 1 January 2020 to 4 May 2020. Systematic review and search of the published literature was done using the Preferred Reporting Items for Systematic Review and Meta-Analyses (PRISMA). (4) Results: In total, 11 case series studies comprising 104 pregnant women with COVID-19 were included in our review. Fever (58.6%) and cough (30.7%) were the most common symptoms. Other symptoms included dyspnea (14.4%), chest discomfort (3.9%), sputum production (1.0%), sore throat (2.9%), and nasal obstruction (1.0%). Fifty-two patients (50.0%) eventually demonstrated abnormal chest CT, and of those with ground glass opacity (GGO), 23 (22.1%) were bilateral and 10 (9.6%) were unilateral. The most common treatment for COVID-19 was administration of antibiotics (25.9%) followed by antivirals (17.3%). Cesarean section was the mode of delivery for half of the women (50.0%), although no information was available for 28.8% of the cases. Regarding obstetrical and neonatal outcomes, fetal distress (13.5%), pre-labor rupture of membranes (9.6%), prematurity (8.7%), fetal death (4.8%), and abortion (2.9%) were reported. There are no positive results of neonatal infection by RT-PCR. (5) Conclusions: Although we have found that pregnancy with COVID-19 has significantly higher maternal mortality ratio compared to that of pregnancy without the disease, the evidence is too weak to state that COVID-19 results in poorer maternal outcome due to multiple factors. The number of COVID-19 pregnancy outcomes was not large enough to draw a conclusion and long-term outcomes are yet to be determined as the pandemic is still unfolding. Active and intensive follow-up is needed in order to provide robust data for future studies.

## 1. Introduction

In December 2019, outbreaks of respiratory disease caused by a novel severe acute respiratory syndrome coronavirus (SARS-CoV-2) were reported in China. The disease was officially named coronavirus disease 19 (COVID-19) by the World Health Organization (WHO) and was declared a pandemic on 11th March 2020 [1]. Infected patients typically present with fever, cough, or fatigue [2]. Ground glass opacities (GGO) are a commonly seen radiological finding [3]. Reported estimates for the case fatality rate range from 3% to 15% [4].

Maternal physiological changes during pregnancy predispose pregnant women to infectious disease. Physiologically, progesterone-mediated vasodilation observed in pregnancy increases mucosal edema and leads to rhinitis or epistaxis. Lung fluid is also increased, and the diaphragm is elevated 4–5 cm to encompass the growing fetus [5]. During the later stages of pregnancy, the activity of cluster of differentiation (CD)4+ T cells, CD8+ T cells, B cells, and natural killer cells are reduced, increasing the pregnant woman’s vulnerability to severe infection [6]. Viral pneumonia is the most common non-obstetric infection in pregnancy [7] and has been historically associated with increased morbidity and mortality in pregnant women [6,8]. One-fifth of pregnant women with pneumonia experience respiratory failure necessitating mechanical ventilation [9]. Furthermore, viral pneumonia is typically more severe and less responsive to treatment compared to bacterial pneumonia among pregnant women [7]. Higher estradiol and progesterone concentrations promote the development of T-helper cell type 2 (Th2) cells, which drive a shift toward humoral, rather than cell-mediated immunity and further reduce the body’s capability to fight viral incursion [6,10,11]. The developing fetus is also highly susceptible to infection, hypoxia, and aberrantly increased maternal cytokines or complement activation associated with infection [12,13]. Pneumonia during pregnancy is associated with low birthweight, preterm birth, cesarean section, preeclampsia/eclampsia, and Apgar scores <7 at 5 min [14].

Previous studies have shown that infection with severe acute respiratory syndrome coronavirus (SARS) or Middle East respiratory syndrome (MERS) during pregnancy is linked to adverse outcomes. For example, increased maternal death, spontaneous abortions, and intensive care unit (ICU) admission in pregnancies with SARS-CoV have been previously reported [15,16,17]. In a case series with 11 pregnant patients with MERS, the case fatality rate was 35% [18]. However, data on the outcome of COVID-19 infection during pregnancy, including prognosis, are scarce. Furthermore, although recent results show that the isolation associated with COVID-19 control measures is not associated with depressive mental outcomes in pregnant women, the physiological health outcomes of pregnant patients of COVID-19 require further study [19]. It is important that the effect of COVID-19 on the health outcomes of pregnant patients is clearly understood so that healthcare workers can be adequately equipped with knowledge about the prognoses and management of pregnant women with COVID-19. Thus, to aggregate growing evidence regarding this topic, we performed a systematic review evaluating the prognosis and other factors, including clinical characteristics, management, and associated health outcomes of COVID-19 infection in pregnancy.

## 2. Materials and Methods

### Literature Search Strategy and Selection Criteria

We performed a comprehensive systematic review and search of the published literature using the Preferred Reporting Items for Systematic Review and Meta-Analyses (PRISMA). PubMed, Scopus, MEDLINE, Google scholar, and Embase were the databases used to search for evidence and articles published between 1 January 2020 and 4 May 2020. Only articles in the English language were included. A combination of “Coronavirus 19”, “COVID-19”, “SARS-CoV-2”, “2019-nCoV”, “novel coronavirus 2019”, “pregnancy” were the search Medical Subject Headings (MeSH) terms (Appendix A). Manual search and screening of all articles were also carried out. Our inclusion criteria consisted of all manuscript discussing pregnant women affected with COVID-19 infection, including obstetrical, perinatal, and neonatal outcomes and complications. All other manuscripts were excluded. We included case series and retrospective studies, excluding simple case reports.

Two investigators (K.H.L. and M.K.(Moonsu Kwak)) appraised and screened the data independently. The examiners made sure not to include any overlap or duplicates. Any discrepancy was resolved via discussion and subsequent consensus. The quality of the studies that satisfied the inclusion criteria was evaluated using the Newcastle–Ottawa scale by two different investigators autonomously. The appraisal framework reported by Murad et al. [20] was used to adjust for the added risk of bias created by case series analysis. The studies were categorized as either of good, fair, or poor quality. Three hundred twenty one articles were manually reviewed by title screening. One hundred eighty two articles were excluded because they did not report any data regarding COVID-19 or COVID-19 in pregnant women. After reviewing 139 articles manually by abstract screening, 40 manuscripts were further excluded because they did not satisfy our inclusion criteria. Out of the remaining 99 articles, 88 were excluded after full text screening. Therefore, 11 articles were eligible for review in our study. A flow-chart of literature search is presented in Figure 1. Statistical analysis was carried out using SPSS (Version 25, IBM). Continuous variables were reported as means, and categorical ones as percentages.

## 3. Results

### 3.1. Summary of Previously Published Meta-Analyses

Until now, seven meta-analyses on adverse pregnancy outcome of COVID-19 have been identified (Table 1). Mascio et al. [21] performed a meta-analysis on pregnancy outcomes of patients with SARS, MERS, and COVID-19. Chest computed tomography (CT) can play an important role as a viable screening tool for COVID-19 in the pregnant patient even in cases without clinical symptoms; however, no significant results including chest CT findings and treatment modalities were described in the manuscript. Four studies—Kasraeian et al. [22], Yang et al. [23], Gatta et al. [24] and Zaigham et al. [25]—summarized the clinical characteristics of pregnant patients and the overall outcomes of pregnancy, not describing CT findings and type of treatment. Martar et al. [26] described overall clinical characteristics, CT findings, and outcomes of pregnancy without type of treatments. Lastly, Abdollahpour et al. [27] described the previous published papers, but there are no summarized data of clinical pregnancy outcomes.

### 3.2. Study Selection and Maternal Characteristics

We identified 321 articles. Ninety-nine articles were reviewed by full text screening, and 11 articles on COVID-19 prognosis in pregnant women were eligible for quantitative analysis. All 11 studies were case series on COVID-19 prognosis in pregnant women. Many articles did not include discussion or did not focus on maternal prognosis in pregnancy with COVID-19. These 11 articles adequately reported maternal prognoses and were included in the final quantitative analysis (Appendix A) [28,29,30,31,32,33,34,35,36,37,38].

These 11 studies encompassed 104 pregnant women infected with COVID-19. Maternal characteristics are summarized in Table 2. Sixty-eight percent of women were between the ages of 25 and 34. The median gestational age (GA) was 37 weeks and 6 days. Ninety-one cases originated from China, and nine cases from Iran and four cases from Italy. Two pregnancies were dichorionic diamniotic twin pregnancies and were each counted as one delivery in analyses. COVID-19 was confirmed by reverse-transcriptase polymerase chain-reaction (RT-PCR) in 72 patients. When RT-PCR was not performed, diagnosis was confirmed by the presence of symptoms and abnormal radiological findings on CT scans. Clinical characteristics of each case are reported in Appendix A.

### 3.3. Maternal Findings and Outcomes

Initial symptoms occurred in 74 patients (71.2%) and are summarized in Table 3. Symptoms most commonly reported were fever (58.6%), cough (30.7%), and dyspnea (14.4%). Less common were chest discomfort (3.9%) and sore throat (2.9%). One patient reported nasal congestion, and another single patient reported excessive sputum production. Forty-seven women reported only one symptom at the time of admission.

Fifty-two patients (50.0%) eventually demonstrated abnormal chest CT, and only one patient did not present with any abnormal findings (Table 4). Of those with GGO, 23 (22.1%) were bilateral and 10 (9.6%) were unilateral. When bilateral or unilateral GGOs were not specified, 23 (22.1%) were positive, and only 1 (1.0%) was negative for GGO.

A majority of women (n = 73, 70.2%) received no treatment (Table 5). Most common treatments reported were antibiotics (25.9%), antivirals (17.3%), and hydroxychloroquine (9.6%). Common antibiotics included ceftriaxone, vancomycin, and azithromycin. Common antivirals included oseltamivir and lopinavir/ritonavir. Twenty women received one or two types of treatment, such as antibiotics combined with antivirals, and eleven women received three or more types of treatment.

Gestational diabetes mellitus, preeclampsia, pre-labor rupture of membranes (PROM), fetal distress, abortion, prematurity, pregnancy-induced hypertension, placental abruption, vaginal bleeding, and fetal death were reported as obstetrical complications (Table 6). At least one obstetrical complication was reported in 46 patients. Out of the total cohort of 104 patients, commonly reported obstetrical complications included fetal distress (13.5%), PROM (9.6%), and prematurity (8.7%). Out of the 30 patients without clinical symptoms for COVID-19, obstetrical complications occurred in 15 cases. Out of the 104 with presenting symptoms, obstetrical complications occurred in 31 cases. Four of the five fetal deaths occurred in patients with severe COVID-19 exacerbation and who required ICU admission prior to delivery.

Survival data were reported for 63 patients (Table 2). Seven maternal deaths with COVID-19 infection were reported by Hantoushzadeh et al. [38] The seven women reported at least fever, dyspnea, and cough no more than seven days before admission and died following rapid decompensation to acute respiratory distress syndrome (ARDS) or septic shock. Five died within ten days of admission, and the longest survival time after admission was 22 days. Two were 40–49 years old, and the remaining cases were 25–39. Specifically, patient 1 was admitted with complaints of sore throat and nasal congestion, in addition to fever, dyspnea, and cough. Twenty-four hours later, she suffered ARDS and died shortly after following acute hypotension and bradycardia. Patient 2 developed acute hypoxemia within 2 days of admission and decompensated to ARDS following a cesarean delivery. Patient 3 decompensated within 36 h of admission and died within the next 3 days while suffering from persistent hypoxia and end-organ failure. Patient 4 was admitted with pneumonia and tachycardia. She acutely decompensated 4 days later and died 5 days after admission. Patient 5 was admitted with pneumonia, which had developed a day after onset of fever and dyspnea. She steadily deteriorated over 5 days and died. Patient 6 presented with a dichorionic diamniotic twin gestation. A day later, she acutely decompensated with ARDS, with concern for secondary bacterial pneumonia. She initially improved but experienced a recurrence of ARDS and died following septic shock, disseminated intravascular coagulopathy, and left heart failure. It was unknown if she was SARS-CoV-2 positive at the time of ARDS recurrence. Patient 7 underwent a cesarean delivery due to intermittent hypoxia on admission day 3, and she was intubated 2 days later. She died 23 days after admission while on ventilator support.

A comparison of clinical characteristics of the mothers who survived against those who did not is shown in Table 7. Mothers who suffered severe illness and death were more likely to present with symptoms at hospitalization, namely, with fever and dyspnea (*p* = 0.021 and 0.000, respectively), abnormal CT findings (*p* = 0.055), and bilateral GGOs (*p* = 0.020). These mothers were also receiving more aggressive treatment, with higher rates of antibiotics, antivirals, anticoagulants, and hydroxychloroquine. Mothers who suffered illness and death were older, though this was below significance (*p* = 0.118).

### 3.4. Neonatal Findings and Outcomes

In concordance with previous findings, vertical SARS-CoV-2 transmission was not found through neonatal SARS-CoV-2 RT-PCR because there were no positive results in all neonates who were tested for SARS-CoV-2 (Table 8). Out of patients with recorded delivery methods, cesarean delivery (52 cases) was more common than normal spontaneous vaginal delivery (NSVD) (22 cases). Neonatal survival outcomes were recorded for 63 cases, and 57 were alive by each study’s conclusion.

In the six cases with fetal or neonatal death, most of the mothers suffered from severe COVID-19 and experienced exacerbation of illness as well as hypoxemia prior to fetal death. Four cases of them were reported by Hantoushzadeh et al. [38] In case 1 (GA 30 + 3 weeks), a stillbirth was delivered by normal spontaneous vaginal delivery less than 24 h after the mother suffered from ARDS. In case 2 (GA 24 + 0 weeks), the fetal death occurred 72 h after admission, after the mother suffered dyspnea with acute hypoxemia. Case 3 (GA 24 + 0 weeks) was a twin gestation that suffered intrauterine death following ARDS, which recurred in the mother 2 weeks after initially stabilizing treatment. Case 4 (GA 36 + 0 weeks) was an intrauterine fetal death upon arrival to labor and delivery, though the mother did not experience hypoxemia or ARDS before delivery.

Fetal death was precipitated by symptom presentation also observed in adverse maternal outcomes (Table 9). Fetal deaths were associated with maternal presentation of cough (*p* = 0.008) and dyspnea (*p* = 0.000). Additionally, the mothers were more likely to be treated aggressively with antibiotics, antivirals, anticoagulants, and hydroxychloroquine.

## 4. Discussion

Compared to SARS and MERS, COVID-19 is even more highly infectious. Evaluating the outcomes and prognosis of COVID-19 during pregnancy is important to control disease severity because adverse outcomes are associated with SARS and MERS in pregnant women [7,14,18]. Neonatal outcomes, surgical and anesthesia management, and the possibility of vertical transmission in pregnant women with COVID-19 have already been reviewed [21,22,23,24,25,26,27]. However, to the best of our knowledge, few groups have evaluated the prognoses of pregnant women infected with COVID-19 [23,27], and none have done so with the most recent evidence. As COVID-19 is a global threat, it is critical that the knowledge base is continuously updated with recent evidence. This review summarizes the findings of 104 pregnancies with COVID-19, with a focus on maternal symptoms, complications, and outcomes, using articles published until 4 May, 2020. Prognosis of COVID-19 with pregnancy is similar to nonpregnant adults with COVID-19, but there remains a possible risk in pregnant women for sudden-onset exacerbation of COVID-19 infection in later trimesters.

Previous systematic reviews have demonstrated that the clinical characteristics of pregnant women with COVID-19 resemble those of nonpregnant patients with COVID-19 [21,22,23,24,25,26,27] (Table 1). In concordance with these reports, few maternal and neonatal deaths were reported in the 104 patients examined. Common initial symptoms were fever, dyspnea, and cough. In 30 cases, patients presented with no clinical symptoms. However, abnormal radiological findings and unilateral or bilateral GGOs were present in all but one case, even in cases without clinical symptoms, suggesting that chest CT may remain a viable screening tool for COVID-19 in the pregnant patient. Common obstetrical complications were PROM, fetal distress, and premature birth, and the presence of clinical symptoms was not associated with obstetrical complications. Cesarean section was the more prominent delivery method, though we report a significant proportion of NSVDs (22 out of 74 specified cases). NSVDs were not associated with poorer outcomes compared to cesarean sections, though indications for cesarean delivery were largely unspecified. Antibiotics, antivirals, and hydroxychloroquine were the most frequently used treatments, and 20 of the 31 who were administered treatment received one or two types of treatment. As nearly all the mothers and neonates survived, whether certain symptoms, obstetrical complications, or treatments were associated with better outcomes could not be determined. Nevertheless, it is important to aggregate these findings to further our understanding of COVID-19 in pregnancy. Particularly concerning treatments, this study aggregates the current treatments for pregnant patients with COVID-19 across multiple treatment centers alongside neonatal outcomes and obstetric complications, which may contribute to further studies.

There was little evidence of vertical transmission of SARS-CoV-2. Out of the present 46 cases in which neonatal RT-PCR results were provided, none showed positive results. In separate studies, Yang et al. reported that two neonates out of 84 live births tested positive by RT-PCR at 36 hours and 3 days of age but also stated that postnatal infection was possible [27]. Wang et al. also reported a case of positive RT-PCR findings in one neonate out of 75, at 36 hours of age that had no contact with the mother after birth [23]. Coupled with some findings of lymphocytopenia, thrombocytopenia, and radiological abnormalities in neonates born to mothers with COVID-19, vertical transmission cannot be ruled out [23].

In all of the cases with maternal death, women decompensated rapidly 1–2 weeks following initial onset of fever, dyspnea, cough, and abnormal radiological findings. This is also concerning for neonatal outcomes, as all but one of the neonatal deaths occurred after maternal progression to ARDS or decompensation. All of the women were within the second to third trimester. The available evidence does not show if any clinical symptoms or radiological findings were predictive of the rapid deterioration in these cases, and it cannot be determined whether later stages of pregnancy are associated with worse outcomes. However, as with nonpregnant adult patients, sudden respiratory decompensation leading to severe disease can occur in pregnant patients [39]. Recently, New York–Presbyterian Allen Hospital and Columbia University Irving Medical Center screened a total of 215 pregnant women delivered infants and identified that most of the patients with confirmed COVID-19 at delivery were asymptomatic and more than one of eight asymptomatic women who were admitted to the delivery room unit were positive for SARS-CoV-2 [40]. Additional future studies not only focusing on the presenting symptoms, laboratory findings, and radiological findings but also asymptomatic pregnant women associated with COVID-19 are warranted and necessary. Furthermore, the possibility of exacerbation of COVID-19 infection during later stages of pregnancy should be further evaluated.

Notably, the degree of symptom presentation at the time of hospitalization was associated with both maternal and neonatal death. Specifically, a higher proportion of women presenting with dyspnea and cough proceeded to suffer severe illness and subsequent maternal and neonatal death. In mothers, bilateral GGOs were also predictive of adverse outcome. Subsequently, women with worse outcomes were administered more intensive treatment. As the mortality cases from Hantoushzadeh et al. all decompensated quickly to ARDS and death, it is of high importance to further evaluate symptoms at hospitalization, particularly cough and dyspnea, as predictors of decompensation [38].

The key limitations of the present review are that the studies evaluated often included small sample sizes, were largely limited to cases in a single country, and had missing case data. To accommodate for missing data in statistical analyses, we grouped missing data and results associated with survival (e.g., maternal or neonatal survival) on the assumption that authors were less likely to report on healthy and uneventful pregnancies. Although this may cause some associating factors of maternal and neonatal death to be missed, we found that this approach was more conservative in identifying factors involved in adverse outcome, as only major associations with maternal and neonatal death were distinguished. Second, in some patients, SARS-CoV-2 was diagnosed by clinical and radiological signs without laboratory confirmation. Third, all of the reports of maternal death originated from a single case series, limiting an unbiased perspective of maternal outcomes. We caution against immediate conclusions as a result but nevertheless urge the further pursuit of evaluating presenting symptoms as risk factors for severe maternal COVID-19 outcome.

## 5. Conclusions

This review has comprehensively and judiciously pooled data and combined the latest evidence concerning COVID-19 and pregnancy, a rapidly evolving situation. Pregnant women are vulnerable to pneumonia and other severe coronavirus infections. Although we have found that pregnancy with COVID-19 has significantly higher maternal mortality ratio compared to that of pregnancy without the disease, the evidence is too weak to state that COVID-19 results in poorer maternal mortality ratio due to multiple factors, including the number of COVID-19 pregnancy outcomes we accumulated being not large enough to draw a conclusion. Long-term outcomes are yet to be determined as the pandemic is still unfolding. Active and intensive follow-up is needed in order to provide robust data for future studies.

## Figures and Tables

**Figure 1 jcm-09-03441-f001:**
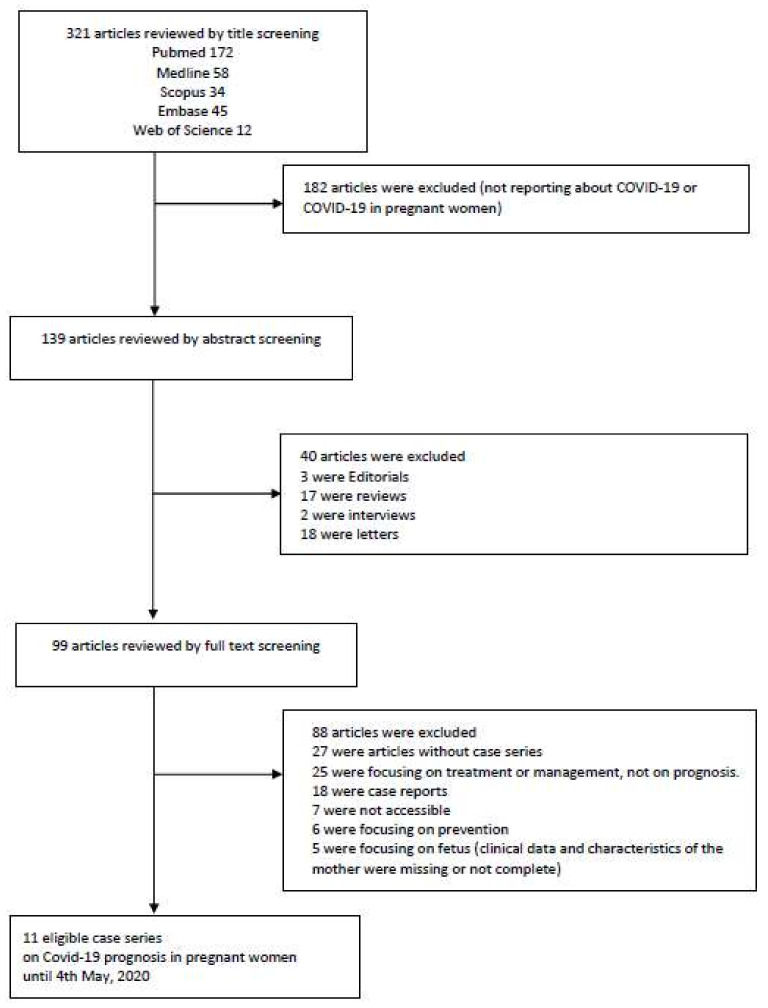
Flow chart of literature search.

**Table 1 jcm-09-03441-t001:** Main characteristics and findings of the previous published systematic reviews/meta-analyses.

Authors	Included Studies, n	Sample Size	Study Period	Description	Comment
Mascio et al. 2020 [21]	19	79	Until 13 March 2020	In mothers infected with coronavirus infections, including COVID-19, >90% of whom also had pneumonia, preterm birth is the most common adverse pregnancy outcome. Miscarriage, preeclampsia, cesarean, and perinatal death (7-11%) were also more common than in the general population.	A meta-analysis of pregnancy outcomes of patients with SARS, MERS and COVID-19—no significant results about CT findings and neonatal outcomes.
Kasraeian et al. 2020 [22]	9	87	Until 18 March 2020	Most pregnant patients suffered from mild or moderate COVID-19 pneumonia with no pregnancy loss, proposing a similar pattern of the clinical characteristics of COVID-19 pneumonia to that of other adult populations.	Summarized the overall outcomes especially including vertical transmission and successful termination—there are no results about CT findings and treatment modalities.
Yang et al. (2020) [23]	18	114	1 January 2020 to 26 March 2020	The clinical characteristics of pregnant women with COVID-19 are similar to those of nonpregnant adults. Fetal and neonatal outcomes appear good in most cases, but available data only include pregnant women infected in their third trimesters.	Summarized the clinical characteristics and neonatal or pregnancy outcomes—no significant results about CT findings and neonatal outcomes.
Gatta et al. (2020) [24]	6	51	Until 14 March 2020	3 pregnancies were ongoing; of the remaining 48 pregnant women, 46 gave birth by cesarean delivery, and 2 gave birth vaginally; in this study, 1 stillbirth and 1 neonatal death were reported.	Summarized the pregnancy outcomes—there are no results about CT findings and treatment modalities.
Zaigham et al. (2020) [25]	18	108	12 Feburary 2020 to 4 April 2020	Most reports described women presenting in the third trimester with fever (68%) and coughing (34%). Lymphocytopenia (59%) with elevated CRP (70%) was observed and 91% of the women were delivered by cesarean section. Three maternal intensive care unit admissions were noted but no maternal deaths. One neonatal death and one intrauterine death were also reported.	Summarized the clinical presentations, laboratory findings and outcomes—there are no results about CT findings and treatment modalities.
Matar et al. (2020) [26]	24	136	Until 30 April 2020	Most common symptoms were fever (62.9%) and cough (36.8%). Laboratory findings included elevated C-Reactive Protein (57%) and lymphocytopenia (50%). GGO was the most common radiological finding (81.7%). Preterm birth rate was 37.7% and cesarean delivery rate was 76%. There was one maternal death. There were two fetal COVID-19 cases.	A meta-analysis of pregnancy outcomes of patients—treatment modalities were not fully described.
Abdollahpour et al. (2020) [27]	29	NA	Until 25 March 2020	In conclusion, improving quality of care in maternal health, as well as educating, training, and supporting healthcare providers in infection management to be prioritized.	Described the previous published papers—there are no summarized data of clinical outcomes.

N: number, COVID-19: coronavirus disease 19, CT: computed tomography, GGO: ground glass opacities, SARS: severe acute respiratory syndrome, MERS: Middle East respiratory syndrome, CRP: C-reactive protein, NA: not mentioned.

**Table 2 jcm-09-03441-t002:** Baseline characteristics of pregnant patients with COVID-19.

Variable	Number of Reported Cases/Total Number of Patients (n = 104)	
**Age (years)**		
<20	0	(0.0%)
20–24	6	(5.8%)
25–29	32	(30.8%)
30–34	38	(36.5%)
35–39	17	(16.3%)
40–45	2	(1.9%)
50–54	1	(1.0%)
No information	8	(7.7%)
**Country**		
China	91	(87.5%)
Italy	4	(3.8%)
Iran	9	(8.7%)
**SARS-CoV-2 Quantitative RT-PCR**		
Positive	72	(69.2%)
Negative	7	(6.7%)
Not done	12	(11.6%)
No information	13	(12.5%)
**Survival Rate**		
Alive	56	(53.9%)
Death	7	(6.7%)
No information	41	(39.4%)

COVID-19: coronavirus disease-19, SARS-CoV-2: severe acute respiratory syndrome coronavirus 2, RT-PCR: reverse transcription polymerase chain reaction, N: number.

**Table 3 jcm-09-03441-t003:** Initial symptoms of pregnant patients with COVID-19.

Initial Symptoms	Number of Reported Cases/Total Number of Patients (n = 104)	
**General Condition**		
Fever	61	(58.6%)
**Respiratory**		
Chest discomfort *	4	(3.9%)
Cough	32	(30.7%)
Dyspnea	15	(14.4%)
Sputum	1	(1.0%)
Sore throat	3	(2.9%)
Nasal obstruction	1	(1.0%)

* Chest discomfort includes chest distress, chest pain, and chest tightness. COVID-19: coronavirus disease-19, N: number.

**Table 4 jcm-09-03441-t004:** CT findings of pregnant patients with COVID-19.

CT Findings	Number of Reported Cases/Total Number of Patients (n = 104)	
**Finding**		
Abnormal	52	(50.0%)
Normal	1	(1.0%)
No information	51	(49.0%)
**GGO ***		
Bilateral	23	(22.1%)
Unilateral	10	(9.6%)
Positive	23	(22.1%)
Negative	1	(1.0%)
No information	47	(45.2%)

* GGO positive is indicated when bilateral or unilateral is not specified in the paper. GGO: ground glass opacity, COVID-19: coronavirus disease-19, CT: computed tomography, N: number.

**Table 5 jcm-09-03441-t005:** Treatment of pregnant patients with COVID-19.

Variable	Number of Reported Cases/Total Number of Patients (n = 104)	
Antibiotics	27	(25.9%)
Antiviral	18	(17.3%)
Steroid	3	(2.9%)
Anti-coagulant	8	(7.7%)
Hydroxychloroquine	10	(9.6%)

COVID-19: coronavirus disease-19, N: number.

**Table 6 jcm-09-03441-t006:** Obstetrical complications of pregnant patients with COVID-19.

Obstetrical Complications	Number of Reported Cases/Total Number of Patients (n = 104)	
Gestational Diabetes Mellitus	3	(2.9%)
Preeclampsia	6	(5.8%)
PROM	10	(9.6%)
Fetal distress	14	(13.5%)
Abortion	3	(2.9%)
Prematurity	9	(8.7%)
Pregnancy induced HTN	4	(3.8%)
Placental abruption	1	(1.0%)
Vaginal bleeding	1	(1.0%)
Fetal death *	5	(4.8%)

* Fetal death includes stillbirth. COVID-19: coronavirus disease-19, PROM; pre-labor rupture of membranes, HTN; hypertension, N: number.

**Table 7 jcm-09-03441-t007:** Evaluation of clinical characteristics on maternal outcome.

Variable		Number of Reported Cases (Total n = 104)		
		Survival or Unreported (n = 97)	Death (n = 7)	*p* Value
Mean age		30	36	0.118
SARS-CoV-2 RT-PCR positive		65 (67.0%)	7 (100%)	0.713
Initial symptoms	Cough	25 (25.8%)	7 (100%)	0.000
	Fever	54 (55.7%)	7 (100%)	0.021
	Dyspnea	8 (8.2%)	7 (100%)	0.000
	Sputum	1 (1.0%)	0 (0%)	0.933
	Sore throat	3 (3.1%)	0 (0%)	0.810
	Nasal obstruction	1 (1.0%)	0 (0%)	0.933
Abnormal CT findings		45 (46.4%)	7 (100%)	0.055
GGO	Bilateral	17 (17.5%)	6 (85.7%)	0.020
	Unilateral	10 (10.3%)	0 (0%)	
	Positive	23 (23.7%)	0 (0%)	
	Negative	1 (1.0%)	0 (0%)	
Treatment	Antibiotics	20 (20.6%)	7 (100%)	0.000
	Antivirals	11 (11.3%)	7 (100%)	0.000
	Steroid	3 (3.1%)	0 (0%)	0.810
	Anticoagulant	2 (2.1%)	6 (85.7%)	0.000
	Hydroxychloroquine	5 (5.2%)	5 (71.4%)	0.000
Delivery method	NSVD	21 (21.6%)	1 (14.3%)	0.003
	C/sec	48 (49.4%)	4 (57.1%)	
Obstetrical complications	Gestational DM	3 (3.1%)	0 (0%)	1.000
	Vaginal bleeding	1 (1.0%)	0 (0%)	0.933
	Preeclampsia	6 (6.2%)	0 (0%)	1.000
	PROM	10 (10.3%)	0 (0%)	0.482
	Fetal distress	14 (14.4%)	0 (0%)	0.352
	Abortion	3 (3.1%)	0 (0%)	1.000
	Prematurity	9 (9.3%)	0 (00%)	0.520
	Pregnancy-induced hypertension	4 (4.1%)	0 (0%)	0.754
	Placental abruption	1 (1.0%)	0 (0%)	0.933
	Fetal death	1 (1.0%)	4 (57.1%)	0.000

SARS-CoV-2: severe acute respiratory syndrome coronavirus 2, RT-PCR: reverse transcription polymerase chain reaction, CT: computed tomography, GGO: ground glass opacities, NSVD: normal spontaneous vaginal delivery, C/sec: cesarean section, DM: diabetes mellitus, PROM: prelabor rupture of membranes, N: number.

**Table 8 jcm-09-03441-t008:** Characteristics of fetal and neonatal outcomes from COVID-19 pregnancies.

Variable	Total Number of Pregnancies (n = 104)	
**Survival Rate**		
Alive	57	(54.8%)
Fetal Death	5	(4.8%)
Neonatal Death	1	(1.0%)
No information	41	(39.4%)
**Delivery Method**		
NSVD	22	(21.2%)
C/sec	52	(50.0%)
No information	30	(28.8%)
**SARS-CoV-2 Quantitative RT-PCR**		
Positive	0	(0.0%)
Negative	46	(44.2%)
Not done	35	(33.6%)
No information	23	(22.2%)

COVID-19: coronavirus disease-19, NSVD: normal spontaneous vaginal delivery, C/sec: cesarean section, SARS-CoV-2: severe acute respiratory syndrome coronavirus 2, RT-PCR: reverse transcription polymerase chain reaction, N: number.

**Table 9 jcm-09-03441-t009:** Evaluation of clinical characteristics on fetal outcome.

Variable		Number of Reported Cases (Total n = 104)		
		Survival or Unreported (n =100)	Death (n = 4)*	*p* Value
Mean age		31	36	0.312
SARS-CoV-2 RT-PCR positive		>0 (0.0%)	0 (0.0%)	1.000
Initial symptoms	Cough	28 (28.0%)	4 (100.0%)	0.008
	Fever	57 (57.0%)	4 (100.0%)	0.140
	Dyspnea	11 (11.0%)	4 (100.0%)	0.000
	Sputum	1 (1.0%)	0 (0.0%)	0.962
	Sore throat	3 (3.0%)	0 (0.0%)	0.888
	Nasal obstruction	1 (1.0%)	0 (0.0%)	0.933
Abnormal CT findings		48 (48.0%)	4 (100.0%)	0.279
GGO	Bilateral	10 (10.0%)	3 (75.0%)	0.301
	Unilateral	10 (10.0%)	0 (0.0%)	
	Positive	23 (23.0%)	0 (0.0%)	
	Negative	1 (1.0%)	0 (0.0%)	
Treatment	Antibiotics	23 (23.0%)	4 (100.0%)	0.004
	Antivirals	14 (14.0%)	4 (100.0%)	0.001
	Steroid	3 (3.0%)	0 (0.0%)	0.888
	Anticoagulant	5 (5.0%)	3 (75.0%)	0.002
	Hydroxychloroquine	7 (7.0%)	3 (75.0%)	0.002
Delivery method	NSVD	21 (21.0%)	1 (25.0%)	0.001
	C/sec	51 (51.0%)	1 (25.0%)	
Obstetrical complications	Gestational DM	3 (3.0%)	0 (0.0%)	1.000
	Vaginal bleeding	1 (1.0%)	0 (0.0%)	0.962
	Preeclampsia	6 (6.0%)	0 (0.0%)	1.000
	PROM	10 (10.0%)	0 (0.0%)	0.663
	Fetal distress	14 (14.0%)	0 (0.0%)	0.556
	Abortion	3 (3.0%)	0 (0.0%)	1.000
	Prematurity	9 (8.0%)	0 (0.0%)	0.692
	Pregnancy-induced hypertension	4 (4.0%)	0 (0.0%)	0.853
	Placental abruption	1 (1.0%)	0 (0.0%)	0.962

* Fetal death in this table excludes stillbirth. N: number, SARS-CoV-2: severe acute respiratory syndrome coronavirus 2, RT-PCR: reverse transcription polymerase chain reaction, CT: computed tomography, GGO: ground glass opacities, NSVD: normal spontaneous vaginal delivery, C/sec: cesarean section, DM: diabetes mellitus, PROM: prelabor rupture of membranes.

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
