# Peer review of "COVID-19 and Adverse Pregnancy Outcome: A Systematic Review of 104 Cases"

_jcm, 2020, doi:10.3390/jcm9113441_

Round 1
Reviewer 1 Report
Thank you very much for your revision. I think this manuscript is ready for publication.
Author Response
Author’s Response to the Comments of Reviewer #1
We sincerely thank reviewer 1 for his or her valuable comments and time in consideration of our manuscript.
COMMENTS FOR THE AUTHOR:
# Reviewer 1
Thank you very much for your revision. I think this manuscript is ready for publication.
Response: We are very appreciative of for reviewer 1’s careful and important comments. Without your great comments, we could not improve our manuscript. Thank again the reviewer for kindly suggesting us to make more concise description of our paper.

Reviewer 2 Report
The systematic review by Ghayda and colleagues provides an overview on pregnancy outcomes of COVID-19 infected pregnant women. The review is based on pooled data of 11 articles. These case studies include 104 pregnant women with COVID-19. The review will be useful for the management of pregnant women with COVID-19 during the current and ongoing pandemic and for risk assessment and identification of parameters for future evaluation of pregnancies with COVID-19. The manuscript is of high quality. The overview of previously published systematic reviews/metaanalyses provided in Table 1 will be very useful for readers.
The following points should be addressed by the authors:
- Line 78: please add „with“ or revise the sentence accordingly
- Line 80: please add intensive care unit to the abbreviation ICU
- Please increase the font size in Figure 1 to increase readibility
- Figure 1: 5 articles were excluded, because they were focusing on the fetus – does this mean clinical data and characteristics of the mother were missing or not complete?
- Line 128: Please explain briefly, why you consider the CT findings important. It would aid the reader through the manuscript, as in line 263 you comment on the relevance of CT findings. The importance of CT findings is already stated in Table 1 in the comment column.
- Line 219: Table 6 should be Table 8
- Line 218: Which evidence supports SARS-CoV-2 vertical transmission? Here the authors state there is little evidence. What is the little evidence?
- Line 227: Would it be possible to provide the information on the gestational week for the fatal cases?
- Line 245: Please explain briefly what you mean by „more vulnerabel“
Author Response
Author’s Response to the Comments of Reviewer #2
We sincerely thank reviewer 2 for his or her valuable comments and time in consideration of our manuscript.
COMMENTS FOR THE AUTHOR:
# Reviewer 2
The systematic review by Ghayda and colleagues provides an overview on pregnancy outcomes of COVID-19 infected pregnant women. The review is based on pooled data of 11 articles. These case studies include 104 pregnant women with COVID-19. The review will be useful for the management of pregnant women with COVID-19 during the current and ongoing pandemic and for risk assessment and identification of parameters for future evaluation of pregnancies with COVID-19. The manuscript is of high quality. The overview of previously published systematic reviews/metaanalyses provided in Table 1 will be very useful for readers.
Response: We are very appreciative of for reviewer 2’s careful and important comments. Without your great comments, we could not improve our manuscript. Thank again the reviewer for kindly suggesting us to make more concise description of our paper.
The following points should be addressed by the authors:
Line 78: please add „with“ or revise the sentence accordingly
Line 80: please add intensive care unit to the abbreviation ICU
Response for all of above: We really thank reviewer 2 for his or her very important comments. We absolutely agree with the comments. Reflecting your important comment, we have made corrections of the sentences in the revised manuscript as follows:
# Introduction (Page 2)
(…) Previous studies have shown that infection with severe acute respiratory syndrome coronavirus (SARS) or Middle East respiratory syndrome (MERS) during pregnancy is linked to adverse outcomes. For example, increased maternal death, spontaneous abortions, and intensive care unit (ICU) admission in pregnancies with SARS-CoV have been previously reported [15-17].(…)}
Please increase the font size in Figure 1 to increase readability
Figure 1: 5 articles were excluded, because they were focusing on the fetus – does this mean clinical data and characteristics of the mother were missing or not complete?
Response for all of above: We sincerely thank reviewer 2 for important comments. We absolutely agree with the reviewer 2’s comment about this important point of view. “Focusing on the fetus” means clinical data and characteristics of the mother were missing or not complete- your comment is right. We increased the font size in Figure 1 and added the statement into the Figure according to your comment.
Line 128: Please explain briefly, why you consider the CT findings important. It would aid the reader through the manuscript, as in line 263 you comment on the relevance of CT findings. The importance of CT findings is already stated in Table 1 in the comment column.
Response: Thank you very much for your important point of view. We absolutely agree with the reviewer’s comment, and thus we have added sentences explaining the importance of CT findings into the paragraph of the revised manuscript as follows:
# Results (Page 4)
Until now, seven meta-analyses on adverse pregnancy outcome of COVID-19 have been identified (Table 1). Mascio et al. [21] performed a meta-analysis of pregnancy outcomes of patients with Severe Acute Respiratory Syndrome (SARS), MERS and COVID-19. Chest computed tomography (CT) can play an important role as a viable screening tool for COVID-19 in the pregnant patient even in cases without clinical symptoms, however, no significant results including chest CT findings and treatment modalities were described in the manuscript. Four studies - Kasraeian et al. [22], Yang et al. [23], Gatta et al.[24] and Zaigham et al.[25] have summarized the clinical characteristics of pregnant patients and the overall outcomes of pregnancy, not describing CT findings and type of treatment. Martar et al. [26] described overall clinical characteristics, CT findings and outcomes of pregnancy without type of treatments. Lastly, Abdollahpour et al. [27] described the previous published papers, but there is no summarized data of clinical pregnancy outcomes.
Line 218: Which evidence supports SARS-CoV-2 vertical transmission? Here the authors state there is little evidence. What is the little evidence?
Line 219: Table 6 should be Table 8
Response for all of above: We really thank reviewer 2 for his or her very important comments. We are very sorry for the confusion not to describe clearly. To address this part clear, we revised the sentences according to your comments as follows:
# Result (Page 9)
In concordance with previous findings, vertical SARS-CoV-2 transmission was not found through neonatal SARS-CoV-2 RT-PCR because there were no positive results in all neonates who were tested for SARS-CoV-2 (Table 86). little evidence for vertical SARS-CoV-2 transmission was found through neonatal SARS-CoV-2 RT-PCR (Table 6). In all neonates who were tested for SARS-CoV-2, there were no positive results. Out of patients with recorded delivery methods, Cesarean delivery (51 cases) was more common than normal spontaneous vaginal delivery (NSVD) (22 cases). Neonatal survival outcomes were recorded for 46 cases, and 41 were alive by each study's conclusion.
Line 227: Would it be possible to provide the information on the gestational week for the fatal cases?
Response: We thank reviewer 2 very much for again carefully rereading our article. We are very sorry to make you confused because we provided the information on the gestational week for the fatal cases only in the supplementary table, not in the manuscript. Reflecting your important comment, we have made corrections in the revised manuscript removing related paragraph as follows (GA is the abbreviation of gestational week, and the full name is stated in the previous paragraph):
# Results (Page 9-10)
In the 4 cases with neonatal death, most of the mothers suffered severe COVID-19 infection and experienced exacerbation of illness as well as hypoxemia prior to fetal death and 5 cases of them were reported by Hantoushzadeh et al.[40] In case 1 (GA 30+3 weeks), a stillbirth was delivered by normal spontaneous vaginal delivery less than 24 hours after the mother suffered from acute respiratory distress syndrome (ARDS). In case 2 (GA 24+0 weeks), the fetal death occurred 72 hours after admission, after the mother suffered dyspnea with acute hypoxemia. Case 3 (GA 24+0 weeks) was a twin gestation that suffered intrauterine death following ARDS, which recurred in the mother 2 weeks after initially stabilizing treatment. Case 4 (GA 36+0 weeks) was an intrauterine fetal death upon arrival to labor and delivery, though the mother did not experience hypoxemia or ARDS before delivery.
Line 245: Please explain briefly what you mean by „more vulnerabel“
Response: We really thank reviewer 2 for his or her very important comments. We are very sorry for the confusion not to describe clearly. To address this part clear and avoid the confusion, we removed the term “more vulnerable” and revised the related paragraph according to your comments as follows:
# Discussion (Page 10)
Compared to SARS and MERS, COVID-19 is even more highly infectious. Evaluating the outcomes and prognosis of COVID-19 during pregnancy is important to control disease severity because adverse outcomes are associated with SARS and MERS in pregnant women [7, 14, 18]. Because adverse outcomes are associated with SARS and MERS in pregnant women [7, 14, 18], it is suspected that this population may be more vulnerable to COVID-19 infection. Evaluating the outcomes and prognosis of COVID-19 during pregnancy is important to control disease severity in this potentially vulnerable population. Neonatal outcomes, surgical and anesthesia management, and the possibility of vertical transmission in pregnant women with COVID-19 have been already reviewed [21-27]. (…)
